# Bed-Load Collision Sound Filtering through Separation of Pipe Hydrophone Frequency Bands

**Jong-Ho Choi [1] , Kye-Won Jun [1],\* and Chang-Deok Jang [2]**

1   Department of Urban Environment & Disaster Management, Graduate School of Disaster Prevention, Kangwon National University, 346 Joongang-ro, Samcheok-si Gangwon-do 25913, Korea; apt105@kangwon.ac.kr
2   Creation and Development, 346 Joongang-ro, Samcheok-si Gangwon-do 25913, Korea; cnd.jang@gmail.com
*   Correspondence: kwjun@kangwon.ac.kr; Tel.: +82-33-570-6816

**Abstract:** Bed-load discharge of a river can be monitored by indirectly measuring the acoustic pulses generated when the bed load collides with a steel pipe installed on the riverbed (i.e., pipe hydrophone measurement). However, existing methods used for filtering pulses from acoustic signals reflect a combination of bed-load collision frequency bands, thereby limiting characterization capabilities. This study proposes an improved filtering method that separates and efficiently examines frequency bands that are highly correlated with bed-load collision characteristics. Herein, an experimental hydraulic model and bed-load collision sound-measurement system were constructed, and bed-load collision experiments were repeatedly performed for collecting acoustic data using a pipe hydrophone. Fast Fourier Transform analysis was performed on data to select the specific frequency bands and pressures reflecting the bed-load particle size. Furthermore, a bandpass method to examine bed-load collision sounds is also presented herein. Results indicate that in comparison with existing filtering methods, the proposed bandpass method yields higher detection rates under bed-load conditions of low flow rate and small particle size, thereby demonstrating its enhanced effectiveness.

**Keywords:** bed load; pipe hydrophone; laboratory experiment; Fast Fourier Transform; pulse filtering

## 1. Introduction

Mountains cover approximately 64% of the Korean peninsula. Consequently, large amounts of bed load are transported from mountains to rivers. Accurate measurement of the characteristics of these bed loads are important for river-management planning, which is essential for mitigation of floods and ecosystem disruption. Therefore, measuring bed-load discharge is crucial from the viewpoint of creating river-management plans. The most common method for measuring bed-load discharge involves the direct physical measurement using a trap or isokinetic sampler, such as the Helly–Smith or Arnhem-type bed-load transport meters [1]. These methods are widely applied to streams flowing over gravel or sand beds [2–6]; however, they are not easily applicable to actual streams of rushing water. Additionally, direct physical measurement is labor- and cost-intensive, and it does not allow the acquisition of continuous data if sediment transport fluctuates significantly with time [7,8]. Consequently, several studies have been conducted in the United States, Japan, and European countries to develop and commercialize novel measuring instruments that are capable of improving the existing methods of measuring bed-load discharge [9–14].

In 1986, a piezoelectric bed-load impact sensor system was developed in Switzerland to measure the temporal changes in bed-load discharge by applying vibrations generated by a moving bed load to an observational technique [15]. A geophone that improves this method's cumbersome system calibration and expands the range of detectable bed-load particle sizes was subsequently developed,

and its applications are now being expanded in European countries [16]. In 1992, a pipe hydrophone was developed in Japan, and its on-field applications continue to be examined. A hydrophone indirectly estimates the bed-load discharge using an acoustic signal generated by particle collisions with a steel pipe [17]. Active acoustic sensors [18–20] and hydrophones [21,22] are used, and Rickenmann [23,24] provided a comprehensive review of techniques and devices for estimating bed-load discharge.

Indirect estimations of bed-load discharge using principles of sound and vibration have been investigated using indoor water-tank tests and on-field applications. Shinichi et al. [25] reported that the amplitude of acoustic signals generated by sediment–hydrophone collisions is strongly correlated with the product of sediment mass and velocity (i.e., momentum), and the corresponding bed-load discharge is correlated with the number of signals (pulses) having amplitudes exceeding a certain level. Based on this finding, Mizuyama et al. [26] continuously monitored bed-load discharge by installing a stage-discharge observation facility using a pipe hydrophone. To quantify bed-load discharge, Tsutsumi et al. [27] linked a pipe hydrophone with a pit sampler to identify the relationship between the intensity of an acoustic waveform and bed-load discharge calibration. Hida [28] performed a water-channel experiment to calculate bed-load discharge and determine the application range of calculations for each bed-load particle size. Uchida et al. [29] reviewed the acoustic wave attenuation phenomenon caused by multiple bed-load collisions, non-collision effects of particles passing through, and recollision effects of bed-load particles caused by wakes. Tsutsumi et al. [30] installed hydrophones vertically on the waterway wall surface rather than horizontally on the river bed. Later, an experiment was conducted using a set of horizontal and vertical pipe hydrophones to measure the bed-load discharge considering the vertical distribution of the particles.

Recently, studies have been performed to estimate particle-size distribution from bed-load discharges of mixed particle sizes. Mao et al. [31] proposed an empirical model that extracted particle-size information of bed loads based on the amplitude ratio of acoustic data recorded on six channels having different sensitivities. This demonstrated the capability of calibrating pipe hydrophones to particle size and transport intensity. Wyss et al. [32] reported that in comparison with exclusively using amplitude information, frequency and amplitude information of plate geophones could be combined to identify a wider range of particle sizes.

As evidenced by the described research trends, most studies concerning bed-load discharge estimation using pipe hydrophones have primarily involved the identification of the correlation between the number of acoustic pulses and amount of bed-load discharge generated when a collision occurs [33]. However, when a large amount of bed-load discharge is transported, the number of pulses generated can be underestimated owing to overlapping signal waveforms, which in turn, can restrict the upper limit of detectible particle characteristics. This is caused by insufficient noise rejection in the raw signal [34]. Koshiba and Sumi [34] devised an improved pulse-count system that uses Discrete Wavelet Transform (DWT), a signal processing technology, for noise reduction. The presented results showed that the DWT was useful in combination with existing pulse-count systems to reduce the signal overlap and to mitigate pulse saturation. Notably, there are two methods for filtering pulses in acoustic signals. The amplification channel method filters the acoustic signal pulses by amplifying them in multiple steps and subsequently filtering each pulse that exceeds a set amplitude threshold. The other is the threshold-setting method, which composes the threshold in several stages without amplifying the signal, recording the number of pulses filtered for each threshold distinction. Tsutsumi et al. [27] showed that if the discharge is high, it is reasonable to use the relationship between the number of pulses and bed-load discharge in medium amplification channels. Hasegawa et al. [35] compared and examined pulse-filtering methods, suggesting that, when the bed-load discharge is high, it is more effective to record the number of pulses using the threshold-setting method than the amplification channel method. Choi et al. [36] performed a basic study on the threshold-setting method that used particle-size amplitude to filter pulses based on bed-load particle size. However, their method underestimated the detection rate because the thresholds of extant studies were established without sufficient consideration of noise filtering and amplitude.

Therefore, the proposed study introduces a frequency analysis method to distinguish collision sound and noise for reducing the difficulty in effectively distinguishing them using pipe hydrophones. For this purpose, bed-load collision experiments were performed on particles with sizes of 9.53–12.70 mm using a laboratory experiment device equipped with a pipe hydrophone. The bed-load collision sound-filtering method employed a bandpass (BP) method, which interprets bed-load discharge thresholds using pulse number based on a specific frequency band and sound pressure. Furthermore the experimental results of Choi et al. [36] were further analyzed with an improved threshold-setting method and were compared with the experimental results obtained in the proposed study; furthermore, the detection rate between the BP method and existing bed-load collision sound-filtering method was analyzed.

## 2. Bed-load Discharge Estimation Method Using Pipe Hydrophone

### 2.1. Classification of Sediment Transport

Sediments are produced by weathering of the earth's crust. They are eroded from mountains, transported, and deposited into rivers by the action of water or wind. As shown in Figure 1a, sediments can be classified as wash load or bed material load depending on their source. Wash loads are transported by suspension, and mainly comprise silt and clay, whereas bed material loads are transported as either suspension or bed loads mostly comprising sand or larger particles. Given that fine wash-load sediments cannot be accurately calculated, only bed material loads are considered when determining the total sediment transport. Bed material loads can be further divided into bed and suspended load. As shown in Figure 1b, bed loads contain a coarse-grained material that moves on a riverbed through sliding, rolling, and saltation, whereas suspended loads comprise a fine-grained material that moves by suspension in water owing to turbulent diffusion in a flowing stream. Our experimentation is focused on identifying the acoustic characteristics of bed-load sediment transport.

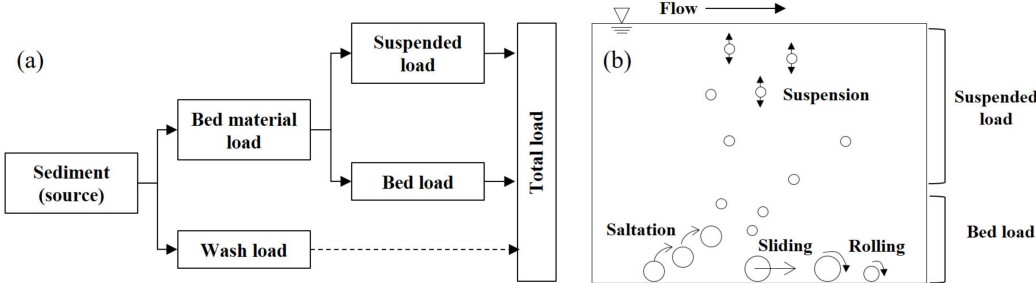

**Figure 1.** Classification of sediment transport. (**a**) Classification of sediment; (**b**) sediment transport modes.

### 2.2. Principles of Bed-Load Discharge Estimation Using Pipe Hydrophone

Pipe hydrophone indirectly measures the bed-load discharge by assessing the acoustic signals of particles when they collide with the pipe. This system is illustrated in Figure 2. A sensor acquires an acoustic signal via the microphone during collision. The waveform is further shaped and amplified; then, a secondary treatment (analysis) is applied to estimate the bed-load characteristics.

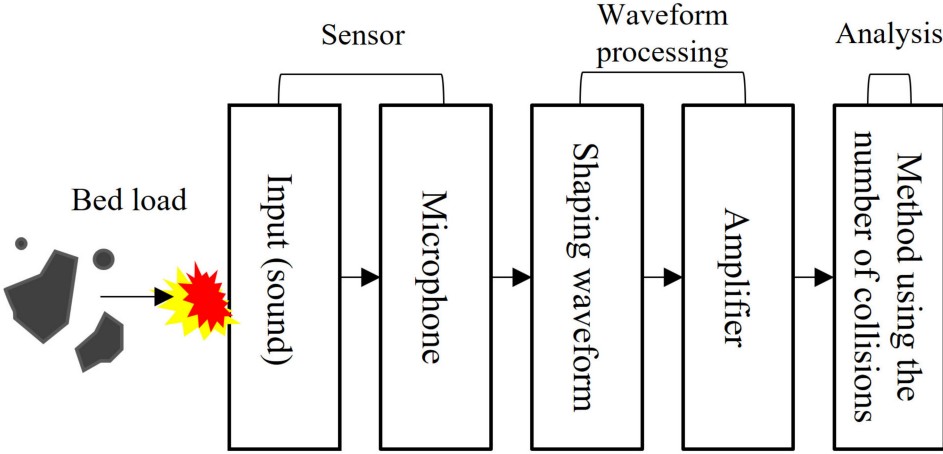

**Figure 2.** Configuration and operational flow diagram of the pipe hydrophone system.

Specifically, the sound signal is detected with a capacitor-type microphone in the pipe and is converted into an electrical signal (sampling rate of 25.6 kHz). The converted raw signal extracts only the positive values, as shown in Figure 3, thereby enveloping the signal peak. The generated enveloped signal is then amplified into six magnifications of 2, 4, 16, 64, 256, and 1024 times. The amplified and raw signals are simultaneously and continuously recorded by the data logger. Methods for estimating bed-load discharge based on the recorded data are introduced in the following sections.

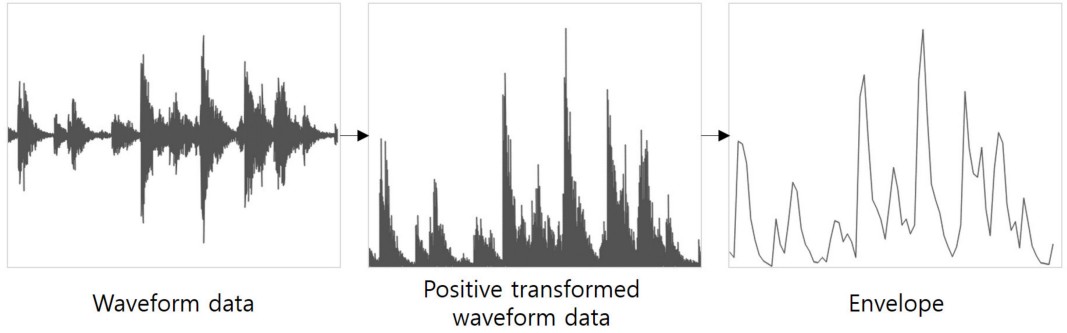

**Figure 3.** Process of converting raw waveform.

### 2.3. Acoustic Signal Filtering and Bed-Load Discharge Estimation Method

There are two filtering methods that set the threshold amplitudes for the acoustic signals measured via the pipe hydrophone for discretizing and passing the pulses so that the particle characteristics can be estimated. The method shown in Figure 4a amplifies the acoustic signals in multiple steps and filters the pulses of each signal that exceed the corresponding threshold [37–39]. Figure 4b presents another method that sets thresholds in multiple steps without amplifying the signals and filters the pulses of each signal that exceed the corresponding threshold [40]. Although two processing methods are used, the same number of pulses is recorded.

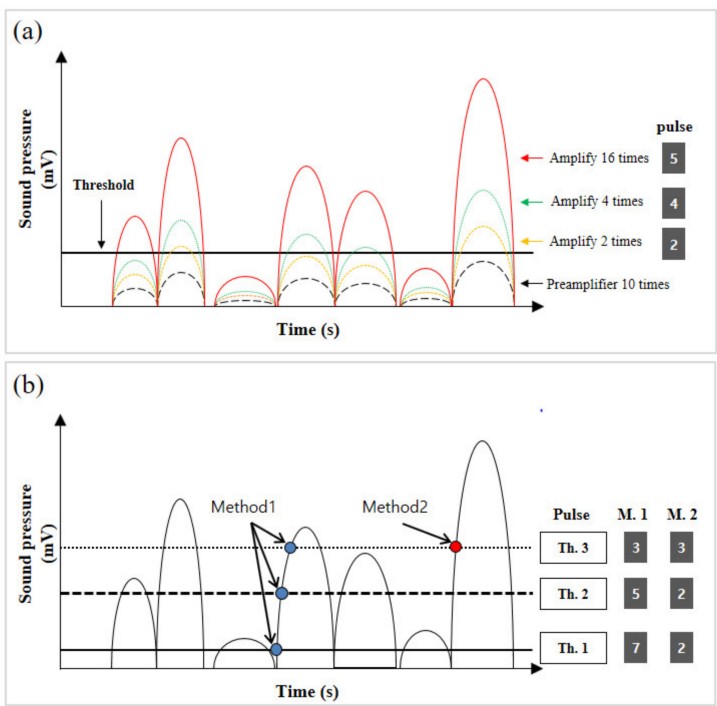

**Figure 4.** Acoustic signal pulse-filtering methods. (**a**) Amplification channel (Method 1); (**b**) threshold setting (Method 2).

There are two pulse-detection methods used for threshold setting. As shown in Figure 4b, the first method records when the shaping signal exceeds a set threshold (Method 1), and the other records the number of pulses having only the maximum threshold (Method 2). The former is evaluated only by the threshold value, where the number of pulses is in focus. The latter is evaluated by the number of pulses detected for each threshold value. Both parts of Figure 4 show the number of pulses recorded from the relationship between signal and threshold. The number of pulses by the amplification channel (Method 1) is measured as follows. When the acoustic signal is amplified by 2, 4, and 16 times based on one threshold, 2, 4, and 5 pulses are detected, respectively. The number of pulses by the threshold setting (Method 2) is as follows. First, the number of pulses detected by Method 1 in the order of threshold values 1, 2, and 3 is 7, 5, and 3, respectively. Second, the number of pulses detected by Method 2 are 2, 2, and 3, respectively.

Based on the pulses obtained by these filtering methods, bed-load discharge is estimated as follows. The amplification channel method considers the capture rates (i.e., number of pulses/particles) of the bed load with diverse particle sizes by selecting a specific amplified signal that best represents the capture rates of particle sizes, among the signals amplified by various factors for estimating the bed-load discharge. The threshold-setting method separates the amplitude threshold value for acoustic signals into multiple stages and estimates bed-load discharge, based on the number of pulses filtered for each threshold of the acoustic signals associated with various particle sizes. Most existing pipe hydrophone systems adopt the amplification channel method. However, this study utilizes Method 2 of the described threshold-setting method, which is known to show better correlation between the number of pulses and bed-load discharge [32] even under large discharges. Furthermore, the threshold for detecting the number of pulses was set based on the positive peak sound pressure appearing in the bed-load collision sound.

*2.4. Bed-Load Analysis via Bandpass (BP) Method*

The BP method was developed by Belleudy et al. [41], who studied the frequency-domain characteristics of particle sizes using a hydrophone. The novel BP method presented in this study

selects a specific frequency band using Fast Fourier Transform (FFT) analysis of the acoustic signals measured by the pipe hydrophone, as shown in Figure 5. This method detects the number of pulses by setting a threshold based on the sound pressure appearing in the corresponding frequency band for each particle diameter. This implementation is an improvement of Method 2 shown in Figure 3. The threshold of Method 2 is set by considering only the sound pressure among the characteristics of the collision acoustics.

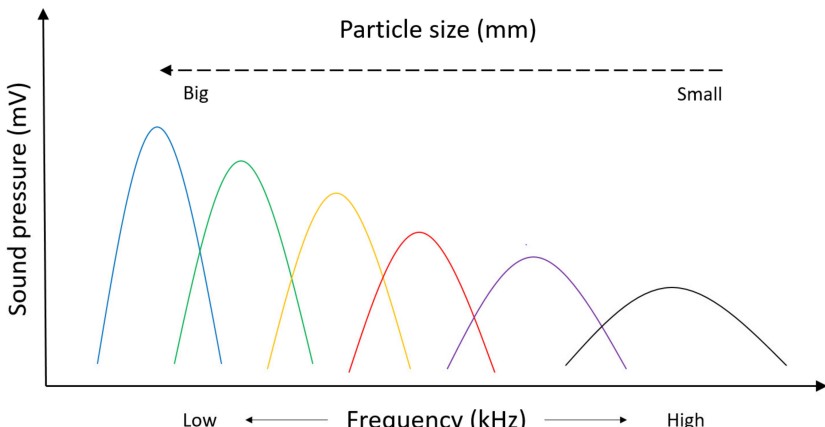

**Figure 5.** Sound-pressure and frequency characteristics by particle sizes.

## 3. Experimental Setup and Method

### 3.1. Experimental Setup

#### 3.1.1. Pipe Hydrophone

Figure 6 presents the specifications and main components of the pipe hydrophone, as applied in the indoor laboratory experiment herein. The components include a stainless-steel pipe with a circular cross section having a length of 33.6 cm, outer diameter of 25 mm, and inner diameter of 20 mm. The stainless-steel box has a length of 60 cm, width of 37.6 cm, and height of 10 cm. The hydrophone is removable and has an integrated structure (Figure 7). To reduce the sound of the bed load colliding with the outside of the circular stainless-steel pipe, the inside of the box is filled with urethane foam. Water and particles flow perpendicular to the longitudinal direction of the pipe hydrophone.

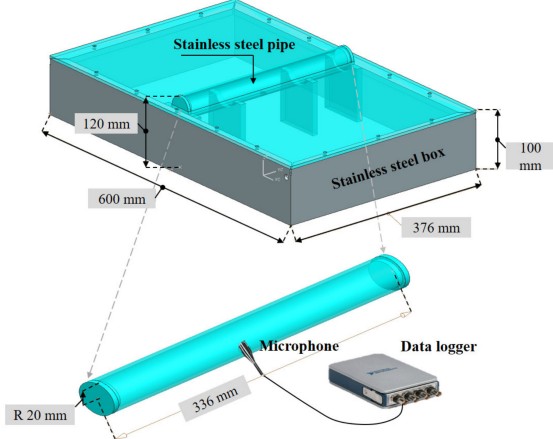

**Figure 6.** Pipe hydrophone main components and specifications. Adapted from [36].

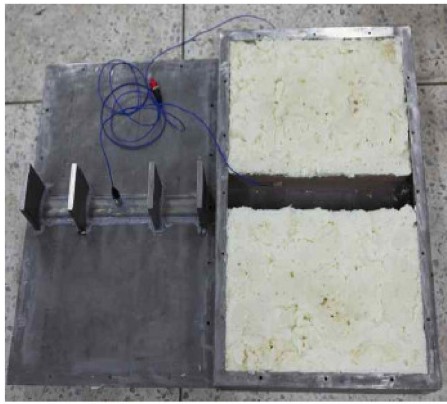

**Figure 7.** Urethane foam internal filling for noise proofing.

The pipe hydrophone has a watertight body, and its completely sealed structure includes a microphone, signal conditioner, and bed-load impact unit, wherein the stainless-steel pipe is exposed to water. The specifications of the microphone, which is installed at the center of the stainless-steel pipe, are listed in Table 1. The microphone is a 130E20 model from PCB Inc. The frequency response is 20–10,000 Hz, and the sensitivity is 39.7 mV/Pa. Acoustic signals detected by the microphone are digitalized in real time using data acquisition and analysis programs and are subsequently processed into data that are effective for measurement. The resultant information is collected by the data logger.

**Table 1.** Pipe microphone specifications.

| Specification | |
| --- | --- |
| Frequency response | 20–20,000 Hz |
| Sensitivity | 39.7 mV/Pa |
| Inherent noise (A weighted) | <30 dB |
| Dynamic range (3% distortion limit) | >122 dB |
| Temperature range | −10–50 °C |
| Excitation voltage | 18–30 VDC |
| Constant current excitation | 2–20 mA |
| Output bias voltage | 5.5–14 VDC |
| Output impedance | <150 Ω |

A database of acoustic data measured by the pipe hydrophone was built using the data logger. For convenient and rapid measurement and analysis, National Instruments (NI) LabVIEW software was used to develop programs for data acquisition and storage. 2016 LabVIEW is a professional development system that implements FFT and windowing using the NI Sound and Vibration Measurement Suite. LabVIEW is a graphical language that can be used to easily develop programs by connecting icons recording the program code. The development program comprises the receiving-channel elements to be measured by the hydrophone and data-expression elements in real time via the time series graph. Additionally, the visualization and evaluation functions of the DIAdem 2017 module were used to analyze the collision sound pressure (Pa) and frequency characteristics (Hz) via the FFT of acquired acoustic data. Data can be sampled at a frequency of 25.6 kHz, enabling the analyses of acoustic signals between 0 and 12.8 kHz according to the Nyquist theory.

3.1.2. Experimental Equipment

Figure 8 shows the frontal view and schematic of the experimental setup. For the experimental channel, an adjustable open channel having a rectangular cross section with a width of 0.4 m, height of 0.4 m, and length of 10 m was used. To address the limitations of the channel's specifications, its width was reduced to 0.2 m for obtaining a higher water level at the same discharge. In the experimental setup with the open channel, water was lifted from a storage tank to an elevated tank using a supply pump.

A triangular weir was operated to allow the water to flow through a lattice screen. Thus, the water flow was stabilized optimally in the channel. The flow rate was adjusted by operating the pump valve. The pipe hydrophone was installed in the lower part of the channel, where the flow of sample particles was stabilized. The observation equipment used to quantify the acoustic characteristics of the bed load consisted of a point gauge (measurement error: ±0.03 mm), which was used to measure the water level, high-speed camera, and video camera to measure the position of the bed-load particles during collision with the pipe hydrophone. The point gauge was installed at the upper point of the pipe hydrophone to measure the central water level, and the video cameras were installed above and beside the pipe hydrophone for observing movement in the longitudinal cross section during collisions. Then, the average sectional flow rate was calculated.

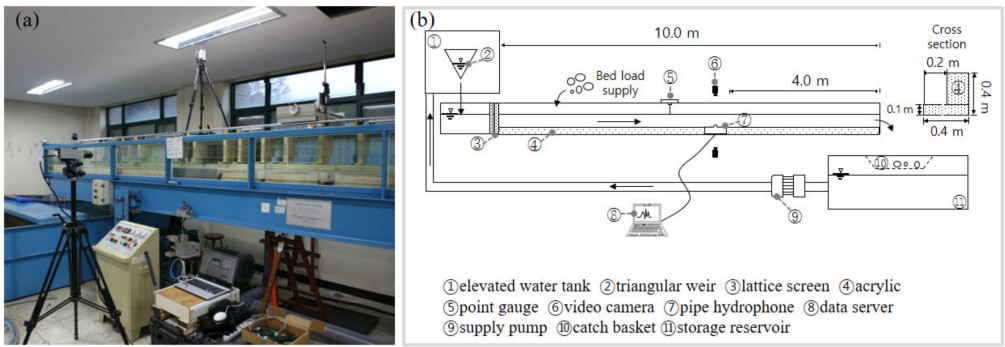

**Figure 8.** (**a**) Frontal view and (**b**) Schematic of experimental equipment.

### 3.1.3. Sediment-Grain Characteristics

To identify the recognition characteristics of the pipe hydrophone for bed-load size, an experiment was conducted using a sieve shaker to permit the flow of two particle sizes; specifically, 9.53 and 12.70 mm. Samples having representative particle sizes were collected from streams in the mountain regions of Hwacheon, Gangwon-do, South Korea. To standardize the physical properties of the samples, the Shape Factor (SF) and specific gravity of each sample were obtained. SF was calculated using the triangular diagrams of Sneed and Folk [42] by extracting 30 samples for each sample, and when the particle approximated a sphere, the shape (s) factor was deemed close to one. An SF ≥ 0.7 can be expressed as a spherical shape, whereas 0.5 ≤ SF < 0.7 is expressed as a transition (t) shape with both spherical and non-spherical characteristics. Finally, an SF < 0.5 is expressed as a non-spherical (n) shape. For each particle size, 30 samples were extracted, and the lengths along the long, short, and intermediate axes were measured (see Table 2).

**Table 2.** Particle shapes of experiment samples.

| Division | | Long (mm) | Intermediate (mm) | Short (mm) | Shape Factor | Sphericity | |
|---|---|---|---|---|---|---|---|
| | | (a) | (b) | (c) | SF=c/√ab | | |
| D (mm) | Average | 12.72 | 9.09 | 5.88 | 0.56 | non-sphere | |
| 9.53 | Standard deviation | 2.73 | 1.69 | 1.42 | 0.13 | s | 6 |
| | | | | | | t | 11 |
| | | | | | | n | 13 |
| D (mm) | Average | 18.85 | 13.7 | 9.17 | 0.58 | transition | |
| 12.70 | Standard deviation | 4.72 | 1.74 | 2.24 | 0.15 | s | 4 |
| | | | | | | t | 15 |
| | | | | | | n | 11 |

The 9.53 mm sample had an SF of 0.56 with a standard deviation of 0.13; therefore, it was non-spherical. The 12.70 mm sample had an SF of 0.58 with a standard deviation of 0.15; therefore, it had a transition shape.

To determine the specific gravity of the samples, the surface dry weight and weight of the sample in 1 L of water were measured. The specific gravity of the 9.53 mm sample was measured to be 2.63, whereas that of the 12.70 mm sample was 2.62.

### 3.2. Experimental Method

For the experimental conditions, the channel slope and flow rate were fixed, and two sample particle types were used. The channel slope was fixed at 0.012, water depth (h) at 0.185 m, and flow rate at 28.15 L/s (see Table 3). The experiment was repeated at least 100 times for each sample particle size, i.e., 9.53 and 12.70 mm. Additionally, the critical bottom velocity and mean critical velocity representing the critical conditions of bed-load particle transport in Table 3 were calculated using Equations (1) and (2), as proposed by Mavis and Laushey [43] and Yang [44], respectively.

$$V_{oc} = 0.153\left(\frac{r_s}{r} - 1\right)^{1/2} D^{4/9} \tag{1}$$

$$V_c = 2.05\omega = 6.508 D^{1/2} \tag{2}$$

where $V_{oc}$ is the critical bottom velocity (m/s), $V_c$ is the mean critical velocity (m/s), $r_s$ is the unit weight of the sediment (kg/m³), $r$ is the unit weight of the fluid (kg/m³), $\omega$ is the final settling velocity (m/s), and $D$ is the diameter of the sediment (mm). In this study, the experimental conditions were selected while considering the mean critical velocity of the particles; it was assumed that they were converted to bed-load discharge by only considering the particles that move along the river bed by the tractive force without considering the fall velocity.

**Table 3.** Experimental conditions.

| D (mm) | Q (L/s) | h (m) | $V_{oc}$; Critical Bottom Velocity (m/s) | $V_c$; Mean Critical Velocity (m/s) |
|---|---|---|---|---|
| 9.53 | 28.15 | 0.185 | 0.54 | 0.66 |
| 12.7 | | | 0.61 | 0.77 |

The experimental procedure is as follows. First, a constant flow rate of 28.15 L/s was supplied inside the experimental channel, after which, the tail gate was adjusted to stabilize the water level and average sectional flow rate. At this point, the average sectional flow rate calculated in the tank was 0.76 m/s. Next, each sample was placed individually into the upper part of the channel to collect acoustic data measured by the pipe hydrophone. These supplied samples approached the pipe hydrophone installed at the lower part of the channel and randomly collided or failed to collide. The collision state was analyzed accordingly using the videos acquired with the high-speed camera and video cameras, and the experimental results were judged as either successes or failures (see Figure 9).

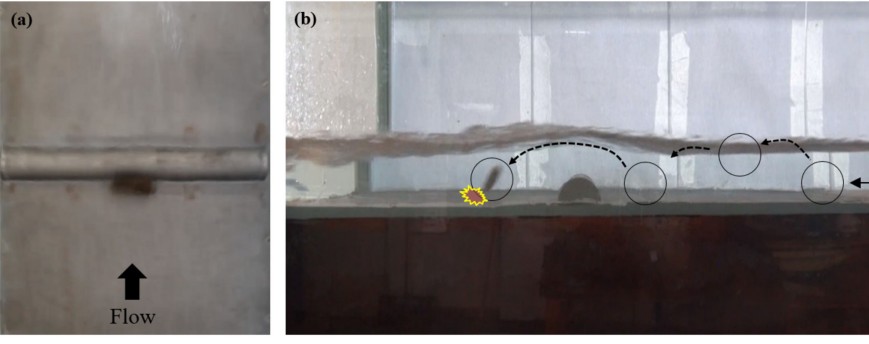

**Figure 9.** Analysis of bed-load collision state using video. (**a**) Success; (**b**) failure.

The bed-load collision sound pressure and video data acquired from the experiment were later used to analyze the characteristics for investigating the bed-load collisions via three methods, i.e., amplification (Method 1), threshold estimation (Method 2), and the proposed BP method.

In the laboratory experiment, the bed-load collision experiment using the pipe hydrophone was performed 140 and 180 times for the 9.53 and 12.70 mm particles, respectively. The failure cases (the individual particles stopped before reaching the pipe hydrophone) and the cases wherein the particles collided with the silicon section bonded to the side wall used to fix the pipe hydrophone were analyzed via video. The number of failure cases for the 9.53 and 12.70 mm particle beds were 35 and 75, respectively, resulting in 105 valid experiment rounds for each bed-load particle size (Table 4).

**Table 4.** Number of particles used in the experiment.

| D (mm) | Number of Runs | Number of Failed Collisions | Number of Collisions with the Pipe Microphone |
|--------|----------------|------------------------------|------------------------------------------------|
| 9.53   | 140            | 35                           | 105                                            |
| 12.7   | 180            | 75                           | 105                                            |

## 4. Laboratory Experiment Results and Analysis

### 4.1. Collision Sound Analysis for Each Bed-Load Particle Size

There are three primary methods of analyzing collision sounds according to the bed-load particle size. These include conventional amplification channel (Method 1), threshold setting (Method 2), and improved BP method. The collision sound thresholds were set for these methods, and the corresponding bed-load collision detection rates were compared and analyzed.

The experimental results previously obtained in [36] were reanalyzed and compared, and the improved performance of the novel BP method was verified. In the previous study, a peak sound-pressure analysis experiment was performed on the collision of five particle sizes (4.75, 9.53, 12.70, 19.05, and 25.40 mm) according to the three-step discharge (14.30, 21.92, and 30.91 L/s) flume changes having a slope of 0.033. In the previous study, the sound pressure used as the reference for the amplification channel had a value of 2 V. Among the six channels (2, 4, 16, 64, 256, and 1024 times), the 256 times channel exhibited the best detection rate. Hence, this channel was selected for comparison to verify the threshold-setting method.

In the previous threshold-setting method [36], it was noted that setting a representative threshold based on the arithmetic mean of the thresholds for each particle size resulted in an underestimated threshold. The improved technique addresses the limitation of the previous method by setting a threshold with a confidence level of 95% according to the normal distribution for each case.

Figure 10 compares the detection rates of the three methods for the previous experimental data, which are listed in Table 5. Herein, the detection rate is the ratio of the number of collision particles and number of pulses. When the average value was used as the threshold, the improved method exhibited an improved detection rate of 11.2% on average at 95% confidence level for each flow rate and particle size. Meanwhile, the amplification channel-detection method yielded a detection rate of 100% for the particle size of 12.70 mm. However, it showed excessive detection for the 19.05–25.40 mm particles. Moreover, small particles (4.75 mm) were not detected, indicating its difficulty in detecting smaller bed-load collision sounds. This characteristic appears to be related to the use of the voltage or amplification channel in the detection method. Depending on the adopted criterion, the detection rate per particle size varies widely. This phenomenon was also observed in the flow rate ranges in all the tested cases.

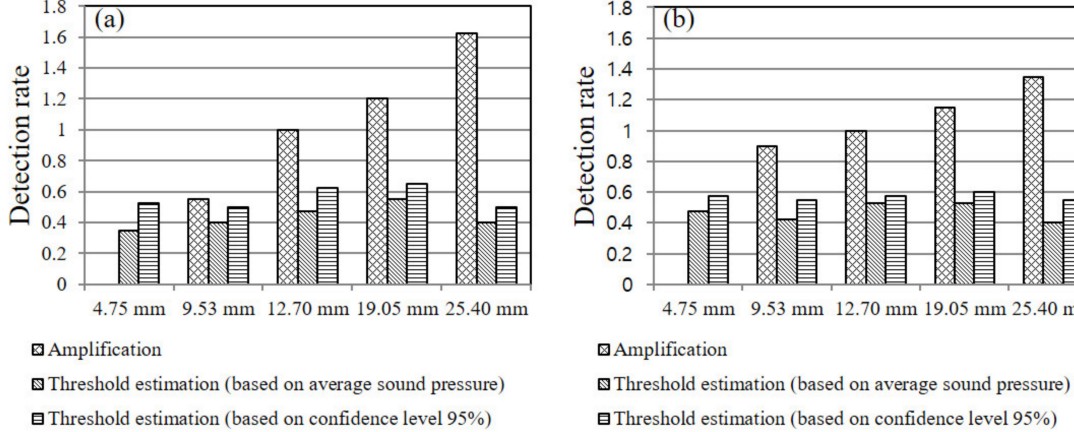

**Figure 10.** Detection rate comparison by filtering method. (**a**) Q = 14.30 L/s; (**b**) Q = 30.91 L/s.

**Table 5.** Comparison of detection rates by acoustic signal filtering method.

| Discharge (L/s) | Filtering Method | Detection Rate | | | | | CV (%) |
|---|---|---|---|---|---|---|---|
| | | 4.75 mm | 9.53 mm | 12.70 mm | 19.05 mm | 25.40 mm | |
| 14.30 | Amplification | 0.00 | 0.55 | 1.00 | 1.20 | 1.63 | 71.26 |
| | Threshold estimation (based on average sound pressure) | 0.35 | 0.40 | 0.48 | 0.55 | 0.40 | 17.99 |
| | Threshold estimation (based on confidence level 95%) | 0.53 | 0.50 | 0.63 | 0.65 | 0.50 | 12.86 |
| 30.91 | Amplification | 0.00 | 0.90 | 1.00 | 1.15 | 1.35 | 59.13 |
| | Threshold estimation (based on average sound pressure) | 0.48 | 0.43 | 0.53 | 0.53 | 0.40 | 12.13 |
| | Threshold estimation (based on confidence level 95%) | 0.58 | 0.55 | 0.58 | 0.60 | 0.55 | 3.67 |

Table 5 lists the threshold calculations of each bed-load collision sound-detection method using existing experimental data. *CV* is the coefficient of variation, calculated by dividing the standard deviation by the arithmetic mean. Values closer to zero indicate a denser distribution of values.

The bed-load collision sound data measured in the experiment were analyzed, and the average sound pressures and deviations for the same particle sizes from the previous experiments were compared, as shown in Table 6. Because the slope was gentler and water level was higher in this experiment, a significantly lower average sectional flow rate was applied. According to the analytical results, significantly low collision sounds of 0.26 and 0.66 mV were generated for the particle sizes of 9.53 and 12.70 mm, respectively.

**Table 6.** Changes of sound pressure by particle size according to hydraulic conditions.

| Slope | Discharge (L/s) | Water Level (cm) | Velocity (m/s) | Sound Pressure (mV) | |
|---|---|---|---|---|---|
| | | | | 9.53 mm | 12.70 mm |
| 0.033 | 14.30 | 2.52 | 1.42 | 8.36 (±1.00) | 17.42 (±2.23) |
| | 30.91 | 4.38 | 1.76 | 13.02 (±2.30) | 26.83 (±3.31) |
| 0.012 | 28.15 | 18.45 | 0.76 | 0.26 (±0.12) | 0.66 (±0.29) |

Table 7 shows the bed-load collision sound-detection rate of each method using the bed-load collision sound data measured in this experiment. The detection rates per amplification channel

were close to 0% for both particle sizes, which are substantially lower than those obtained when the magnitude of the collision sound pressure was high. The detection rate expressed in Table 6 was obtained using the 256 time amplification channel. In this experiment, the improved threshold-setting method yielded better performance for both particle sizes.

**Table 7.** Particle detection rate by filtering method according to change of hydraulic conditions.

| Slope | Discharge (L/s) | Water Level (cm) | Velocity (m/s) | Filtering Method | Detection Rate | |
|---|---|---|---|---|---|---|
| | | | | | 9.53 mm | 12.70 mm |
| 0.012 | 28.15 | 18.45 | 0.76 | Amplification | 0.00 | 0.05 |
| | | | | Threshold estimation (based on average sound pressure) | 0.33 | 0.37 |
| | | | | Threshold estimation (based on confidence level 95%) | 0.41 | 0.48 |

*4.2. Collision Sound Analysis by Frequency Band*

4.2.1. Representative Collision Sound Frequency-Band Analysis by Particle

The frequency of the bed-load collision sound reacted differently according to various environmental factors. This study aims to analyze the frequency bands of collision sounds generated by direct collisions with the fabricated pipe hydrophone and select a specific band. Thus, we present a novel BP method that more accurately detects whether a bed-load collision has occurred and can effectively convert the number of pulses to a specific bed-load discharge. First, the sound generated by flowing water and that generated when the bed load approaches the stainless-steel pipe of the hydrophone were analyzed. The characteristics of the analyzed sound were classified as ambient noise to distinguish it from the sound generated when the bed load directly collides with the stainless-steel pipe of the hydrophone.

For the frequency analysis, the collected bed-load collision sounds were separated into 2-s regions, and the FFT results were analyzed. The spectrum having only water flow before collision and the spectrum obtained during collision were compared to determine the collision frequency region.

Figure 11 shows the time series of sound pressures when only water flows, and the sound-pressure distribution by the frequency band over 4 s when the bed load did not collide.

In Figure 11a, the maximum sound pressure when only water flows is approximately 0.0103 Pa. In Figure 11b, the maximum sound pressure when the bed load approaches the pipe but does not collide is 0.0133 Pa. The value in the case without collision was slightly higher than when only water flows.

Figure 12 shows the sound pressure and corresponding spectrum of a pipe hydrophone when colliding with bed load having a particle size of 9.53 mm.

The sound pressure measured by the pipe hydrophone captured the following movements. As the bed load approached the pipe, the low sound pressure of the contact with the acrylic floor was discretely distributed, and as the bed load reached the pipe hydrophone, it collided several times with the floor of the pipe hydrophone. Then, it collided with the stainless-steel pipe of the hydrophone. In the graphs shown in Figure 12, the dense sound pressure was distributed in the 5–6 kHz region, and the high sound pressure was discretely distributed near 10 kHz. The same patterns were observed in the measured bed-load collision sounds.

However, measurement data of bands near 10 kHz were located closer to 12.8 kHz, suggesting that errors might arise in measurements returned from higher bands. Thus, this study performed analysis based on the 5–6 kHz range and used the results to calculate the threshold for distinguishing bed-load collision sounds.

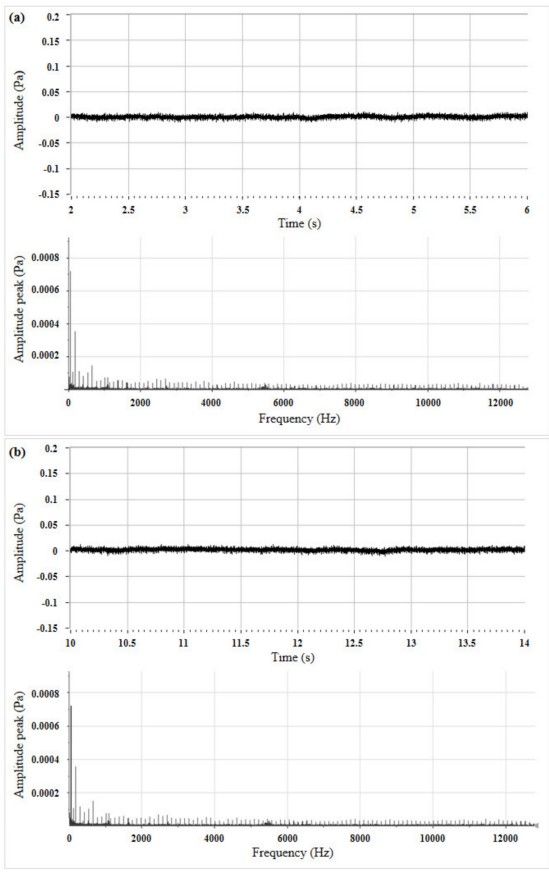

**Figure 11.** Sound pressure and frequency. (**a**) Water only; (**b**) failure case.

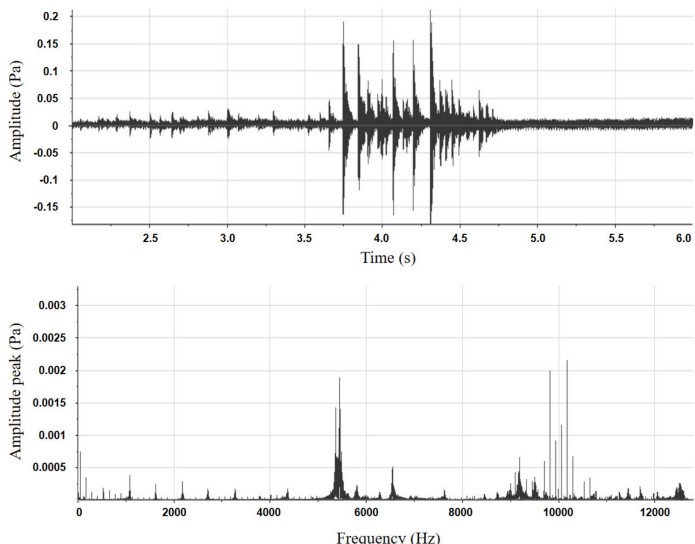

**Figure 12.** Collision sound pressure and frequency for 9.53-mm bed-load particle size.

In Figure 13a, the frequency of the 9.53-mm particle experiment was divided into 1-kHz intervals, and the summed sound pressures were expressed as a percentage of the overall mean. The bars indicated by red diagonal lines represent the largest sound-pressure values. In this study, excluding the 9–10 kHz interval characteristic, the frequency range was specified for the 5–6 kHz intervals. This is because, based on the review of previous literature, the range below 10 kHz was presented as the characteristic frequency band for particle sizes near 10 mm, similar to that used in this study.

Figure 13b shows the averages of the peak sound pressures in each 1-kHz interval of the 9.53-mm particle experiment. Other than cases in which the values of the 1-kHz range were combined, the peak sound pressure was much less than 1 kHz. This can be attributed to the system error, in which high sound pressure was initially measured in the low range near 0 kHz.

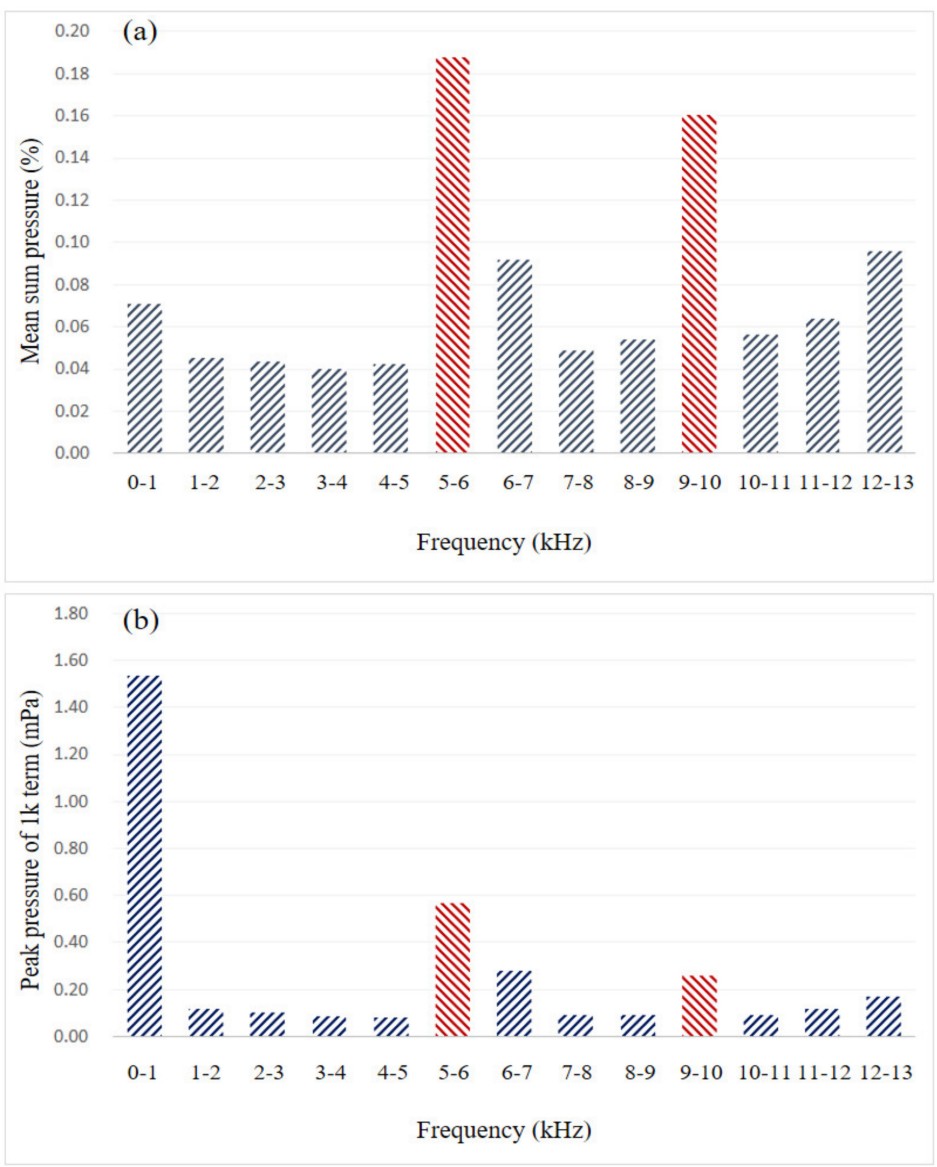

**Figure 13.** Sound pressure in 1-kHz increments with 9.53-mm particles. (**a**) Mean sum; (**b**) peak pressure.

In Figure 14a, the frequency of the 12.70-mm particle-size experiment was divided into 1-kHz intervals, and the summed sound pressures were expressed as a percentage of the overall mean. As in the 9.53-mm experiment, the distribution of sound pressure was most prominent in the 5–6 and 9–10 kHz frequency ranges. Furthermore, the same peak sound-pressure trends were observed as those of the 9.53-mm particle-size experiment (Figure 14b).

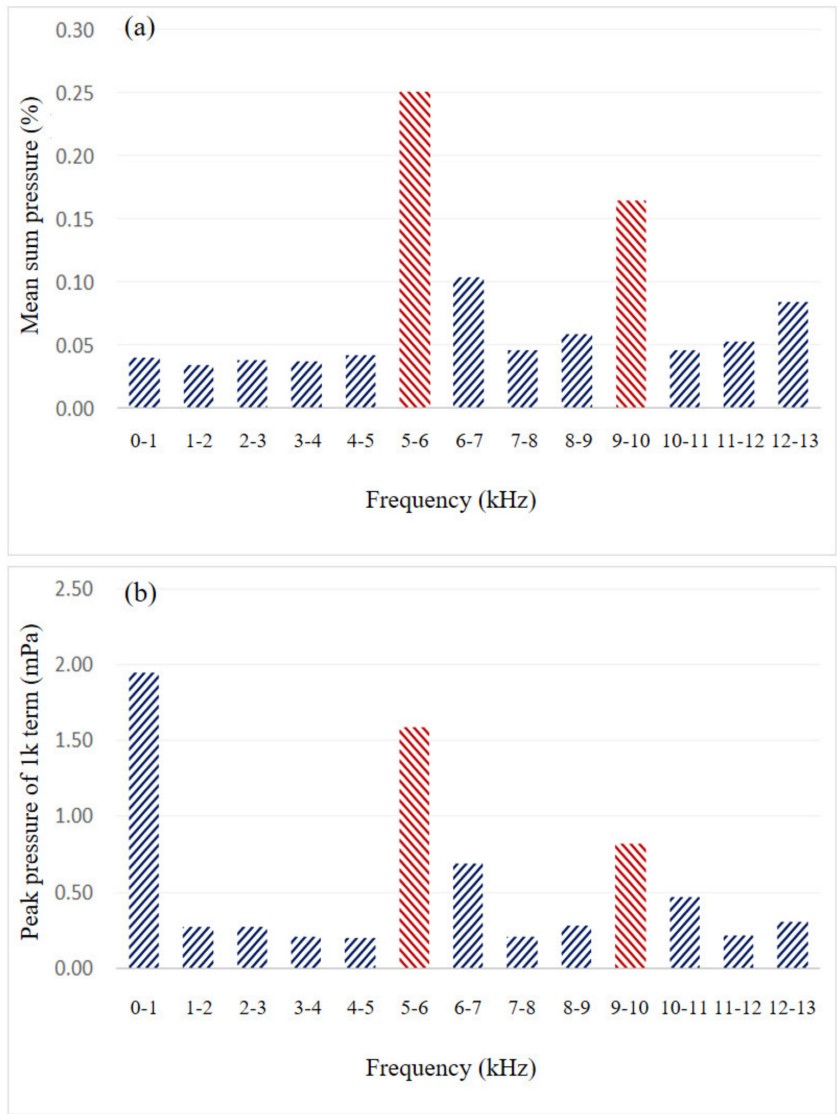

**Figure 14.** Sound pressure in 1-kHz increments with 12.70-mm particles. (**a**) Mean sum; (**b**) peak pressure.

In the frequency range of 5–6 kHz presented herein, the sound-pressure threshold was calculated as 0.181 and 0.417 Pa for 9.53 and 12.70 mm, respectively, and the detection rates for bed-load collision sounds were assessed based on these criteria.

4.2.2. Detection Rate Analysis Result by Bandpass Method

With the BP method, the sound-pressure threshold in the frequency range of 5–6 kHz was calculated as 0.181 and 0.417 Pa for 9.53 and 12.70 mm, respectively. For a particle size of 9.53 mm, the sum of sound-pressure values in the 5–6 kHz range comprised approximately 18.8% of the sound-pressure values in all ranges (Figure 13a). For 12.70 mm, it comprised 25.0% (Figure 14a). Table 8 and Figure 15 show the detection rate from this study's bed-load collision sound-measurement data for each method. The BP method yielded the best detection rate, followed by the improved threshold-setting method, existing threshold-setting method, and amplification channel method. Specifically, the BP method outperformed the improved threshold-setting method by 46% and 42% for the 9.53- and 12.70-mm particle sizes, respectively. However, these results were achieved under extremely limited experimental conditions (i.e., low discharge and small particle size of 12.70 mm or less), and continuous bed-load collision was not examined. The collision sound of the bed load comprising group particles was

different from the characteristics of the collision sound of the single-particle bed load because signal attenuation occurred owing to the interference between particles. Accordingly, if the BP method was reviewed and supplemented with the characteristics of the collision sound occurring on continuous impact on the fast velocity and various particle size conditions, the bed-load discharge estimation method could be considered more feasibly. Similar to this study, Marineau et al. [14] summarized the acoustic frequency bands by bed-load particle size and presented a method for applying them to actual rivers. Further, high correlation between the two factors was confirmed by comparing the measured bed-load discharge to sediment-generated noise measured by hydrophone in Trinity River, California, USA.

**Table 8.** Particle detection rate by filtering method.

| Slope | Discharge (L/s) | Water Level (cm) | Velocity (m/s) | Filtering Method | Detection Rate | |
|---|---|---|---|---|---|---|
| | | | | | 9.53 mm | 12.70 mm |
| 0.012 | 28.15 | 18.45 | 0.76 | By amplification | 0.00 | 0.05 |
| | | | | Threshold estimation (based on average sound pressure) | 0.33 | 0.37 |
| | | | | Threshold estimation (based on confidence level 95%) | 0.41 | 0.48 |
| | | | | Bandpass | 0.87 | 0.90 |

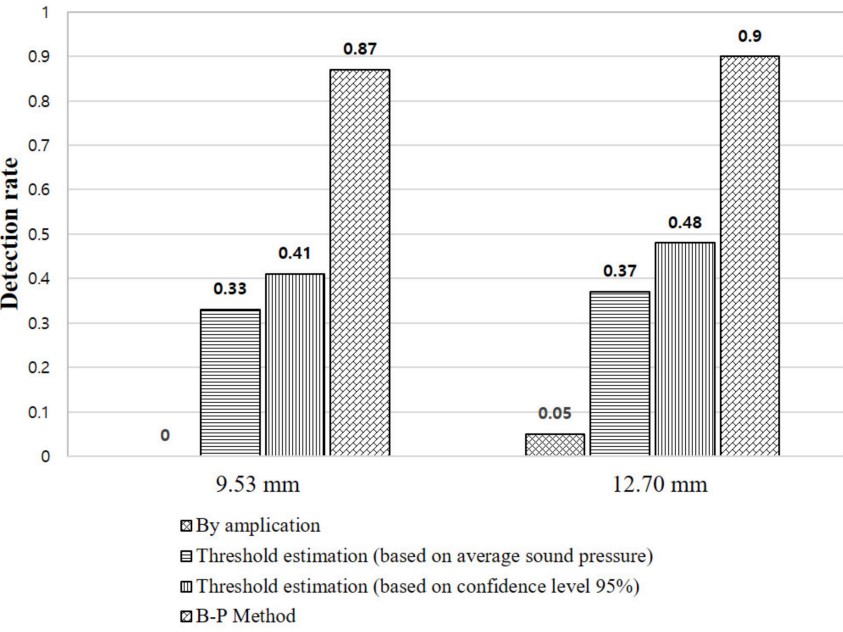

**Figure 15.** Particle detection rate by filtering method.

## 5. Conclusions

In this study, we conducted laboratory experiments using a pipe hydrophone that could continuously measure the bed-load discharge, and also constructed a pipe hydrophone bed-load collision sound-measurement system that facilitated data collection and storage. By measuring and analyzing the collision acoustics of the pipe hydrophone according to the movement of the two bed-load particle types, we compared and analyzed the sound-detection characteristics of the pipe hydrophone according to the existing method, improved method, and proposed BP method. The following conclusions were drawn from this research:

1. The conventional method that measures pulses above the threshold for the acoustic amplification channel could not detect particles of 4.75-mm size. Furthermore, this method resulted in excessive detection for bed-load particle sizes above 12.70 mm. Conversely, although the threshold-setting method exhibited consistent detection overall, the detection rate was considerably low with an average of 45.5%.

2. Setting the threshold for individual particles having a confidence level of 95% enhanced the bed-load collision sound-detection rate by 11.2% in comparison with the previously reported improved method.

3. The detection rate of the amplification channel method was substantially lower than when the magnitude of the collision sound pressure was high.

4. Under the conditions of this experiment, the sound-pressure distribution was observed to be concentrated in the frequency range of 5–6 kHz.

5. By the proposed BP method, the sound-pressure threshold in the frequency range of 5–6 kHz was calculated at 0.181 and 0.417 Pa for 9.53 and 12.70 mm bed-load particle sizes, respectively. The sum of the sound-pressure values of the 12.70-mm particles in the 5–6 kHz range was 33% higher than that of the 9.53 mm particles in the same frequency range.

6. The BP method detection rate for sound pressure in the 5–6 kHz frequency band was calculated at 87% and 90% for particle sizes of 9.53 and 12.70 mm, respectively, which showed better performance than the existing filtering method. In particular, the proposed BP method yielded better detection rates than the other methods under bed-load conditions of low flow rate and small particle size. The proposed method is effective for measuring bed load with low flow rates and small particle sizes.

The proposed BP method improved the bed-load observation technology via the use of pipe hydrophones. However, it is noteworthy that experimental conditions considered herein this study were limited. In the future, the authors plan to enhance practical utilization of the proposed method by performing additional experiments considering different particle sizes and hydraulic conditions.

**Author Contributions:** C.-D.J. developed and manufactured the experimental device for measuring bed-load discharge in mountain streams. J.-H.C. performed experimental data collection and analysis and wrote the manuscript. K.-W.J. oversaw the results of the experiment and revised the manuscript. All authors contributed to the writing of this manuscript and approved the final article. All authors have read and agreed to the published version of the manuscript.

**Funding:** This work was supported by the Basic Science Research Program through the National Research Foundation of Korea (NRF) funded by the Ministry of Education (NRF-2016R1D1A3B03933362), and Supported by the Ministry of the Interior and Safety's National Disaster Management Research Institute (2020-MOIS32-032-00000000-2020).

**Conflicts of Interest:** The authors declare no conflict of interest.

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
