# Peer review of "Bed-Load Collision Sound Filtering through Separation of Pipe Hydrophone Frequency Bands"

_water, doi:10.3390/w12071875_

Round 1
Reviewer 1 Report
I find the content of the manuscript interesting for both the theory and practice of hydrological research.
There are only four small recommendations for improving the perception of the manuscript content:
1) It is advisable to update the list of references (and Introduction) and supplement it with more modern works.
2) Section 4 can be divided into two sections: 4. Results, and 5) Discussion.
3) In Conclusions, please give a more specific description of the shortcomings of your proposed method(s) and suggested ways to overcome them.
4) Some figures need to be made more readable (Figures 7, 10, and 11).
Reviewer 2 Report
This is a review of a paper entitled « Bedload collision sound filtering through separation of pipe hydrophone frequency bands ». The authors reports experiments that were made with a microphone pipe to monitor bedload flux. My general comment on this paper points out the lack of accuracy overall the manuscript.
The structure of the manuscript is fuzzy. For example, there is a global introduction on bedload measurement using indirect methods. And then, in the second part, an other kind of introduction with very basic knowledge. Personnaly, I recommend to delete the second part as most of the material is not usefull for this study. At the end of the introduction, you should arrive to the scientific questions raised with the bibliography and explain how you will study these scientific questions.
Secondly, methods are not well presented. First of all, the processing of the acoustic signal is not enoudh detailed. It is impossible to reproduce the computations with the materials indicated in the manuscript. Indeed, only schematic information (a drawing, fig. 3) are provided. The authors must be more accurate in the description of the methods, particularly concerning the signal processing methods. Every variable has to be defined (with unities, equations).
Concerning the experimental setup, it is well presented but some figures are not usefull (for exsample, fig. 7). Additionally, some runs are presented in the method section but are not used in the result section, and inversely. Again, the presentation of this work should be more accurate.
Finally, I recommend to the authors to strongly revise their manuscript before new submission. Some detailed remarks are available below but are not extensive.
Introduction
The introduction has to be reorganised : 1)review of the literature, 2)questions and 3)how do you raised these questions in this paper.
L50-l70 : In this paragaph, hydrophone/pipe hydrophone/plate geophones are compared as if it was comparable. Concerning pipe hydrophone and plate geophone, I agree that these technologies are similar : measurement of the vibrations of an object impacted by bedload particles. However, the technique of Marineau et al. [14] is very different as it uses bedload self genetared noise : measurements without any object on the river bed. As your paper is dedicated on pipe hydrophone systems, please consider to focus on this technology and only on this technology.
L71-l88 : Correlation between number of acoustic pulses and bedload discharge
- At large bedload flux ~ number of pulses underestimated [overlapp]
- FFT is not a solution for overlapp but for reduction of noise.
Also, I don’t understand your text, I think that two of your statement are opposing :
- L73-75 « However, when a large amount of bed-load discharge is transported, the number of pulses generated may be underestimated »
- L82-84 « Based on a comparison and review of different pulse-filtering methods, Hasegawa et al. [30] observed that the threshold-setting method is effective at recording the number of pulses in the event of large bed-load discharges. »
- So finally ?
Basic Theory
Delete 2.1. Too basic knowledge.
2.2. What is the design of your system – Type of sensor (brand-name ?)? Amplification ? cutoff frequencies of you band-pass filter ? Sampling frequency ?
2.3. By looking at your figure. The main difference between the different methods is not clearly explained. Indeed, in the amplification channel methods, one pulse can belong to several amplification stage. In the threshold-setting method, one pulse belong to one class (betwwen two threshold).
More over, the figure 3 is not sufficient to explain the processing your data. You are recording acoustic wave form. How-do you process the signal. Are the temporal wave form sampled into constant time step (for example, every second is analysed) and then do you take a maximum every second? Do you compute an enveloppe and count every values that are above the threshold ? Do you count the number of sample above the threshold ? The method used to process the signals has to be clearly explained in detailed. A schematic drawing of your processing is not sufficient. You should, for example, use equation to explain the different methods.
2.4. Very basic knowledge. Should be removed from your paper.
2.5.Review literature and not method. Please delete the figure 4 – not usefull to show this figure here. Keep the text but in the introduction.
Experimental Setup and Method
L187 – Which type of hydrophone (brand ? model ? typical frequency response and sensitivity ?)
L193 / Fig 5 – Microphone or hydrophone ?? Please choose and correct over the entire paper
Table 1 – I have my response, it is a microphone. Please use the term of pipe microphone. Delete hydrophone every where.
L207-208 – Please reformulate to read something like : « Data can be sampled at a frequency of 25.6 kHz, enabling the analyses of the acoustic signals betrween 0 and 12.8 kHz according to the Nyquist theory. »
L208-212 /Figure 7 – « LabVIEW 208 is a graphical language that can be used to easily develop programs by connecting icons recording 209 the program code. The development program consists of the elements of the receiving channel to be 210 measured by the hydrophone and data-expression elements that show data characteristics measured 211 in real time through a time-series graph. Figure 7 shows the code in the development program and 212 the user interface. 213
214 « Please delete and also figure 7, not usefull to understand your study.
L248-250 : What is a surface dry weigth ???? why do you measure the weight of the sample with 1L of water ???? A density of 0.4 is very low compare to usual densitiy (around 2.6).
L255 : better to express the slope with usual unities like % [1/82 ~ 1.2 %]
Table 3 : what is this critical velocity ??? not defined in the manuscript.
Result
Part 4.1 – Should be included in the method section.
L275 : you state that the eperiment was performed 140 times for 9.53 mm particle and 180 times for 12.7 mm. So delete the approximate sentence line 256-257 « The experiment was repeated at least 100 times ….. ».
4.1-4.2 : maybe include a table ?
|
D (mm) |
Number of run |
Number of failed collision |
Number of collisions on the pipe-microphone |
|
|
9.53 |
140 |
35 |
105 |
|
|
12.70 |
180 |
75 |
105 |
|
L284 – You state that three methods were employed. However, only two methods were described earlier in the method section. The improved threshold-setting methods has not been presented earlier. The improved method should be presented in details (= with equations) in the method section !
L294-296 – it is not clear for me what is the improved technique. Please develop and explains how are setted the threshold.
Table 4 – Why do you have so many diameters ?? You never mentioned other particles that particles of 9.53 and 12.70 mm. I don’t understand how this table was build. Revise the explanation.
Table 4 – how do you compute the detection rate ??????? Detection rate has not been defined in your manuscript. Please precise.
Table 5 – In the text you defined an average sound pressure. What is the sound pressure reported in your table ? the average sound pressure ? An average sound pressure should be close to zero (as you are analysing wave fluctuations). You should use a root mean square pressure. But if you use a root mean square pressure, you should define a constant temporal window. How do you define the temporal window to compute the root mean square pressure ?? So, meybe the better is to compute the energy (=integral of squared pressure over time) for each impact.
Table 5- In this table, you show different experiments with varying slope and discharge. However, these experiments were not used to test the different methods of signal processing. Why do you present these runs if they are not processed with impacts?
Figure 10- maybe keep the same y-scale as the one of the figure 11 (to compare the signal values by eye).
Reviewer 3 Report
Please, carefully consider the comments in the attached file.

Reviewer 4 Report
Specific comments-questions (to be clarified in the text):
- Equations (1) and (2): Do both, n and k, denote frequencies?
- Figures 5, 7, 10 and 11: The letters and numbers are too small.
- Figure 8b: The numbers within circles are not clear.
- Table 2: (a) What exactly is the column c/a? The shape factor is by SF symbolized. (b) Which is the equation for sphericity? What do s, t and n mean?
- Line 248: Which is the "surface dry weight"?
- Line 289: What does the symbol x mean? (2x, 4x etc.)
- Figure 10: What does "amplitude peak" mean? Max value of amplitude?
- Figures 12b and 13b: (a) What does "1 k term" mean? (ordinate axis). (b) Why is not referred the peak pressure in the interval 0-1 kHZ as the max value?
- A practical result is missing in this study, i.e. a diagram for the bed-load discharge as function of e.g. sound pressure or acoustic pulses frequency.
- See annotated manuscript!

Round 2
Reviewer 3 Report
I acknowledge the improvements made to the original manuscript. However, my previous comments were only partially addressed in the revised manuscript. Please, address all my remaining comments listed in the attached file.

Reviewer 4 Report
Specific comments and questions (to be clarified in the text):
- The old section 2.1 "Classification of Sediment Transport" is absolutely necessary in the revised form of the article, because it is an introductory section for the theme of sediment transport and bed load.
- Table 2: What do the numbers 6, 11 and 13 for s, t and n, respectively, express? The same question is valid for the numbers 4, 5 and 11.
- Table 2: The symbol D (not Dm) should be defined.
- Lines 243 and 244: Is the specific gravity 2.63 or 2.62 dimensionless? How is the specific gravity defined?
- Line 248: The channel slope is 1/82 or 0.012 or 1.2%.
- Equation (1): What are rs and r? Is Voc a flow (water) velocity or particle velocity?
- Equation (2): Vc is a mean (critical) flow velocity.
- Lines 259-261: How is the critical velocity of the particles converted into bed-load discharge?
- Table 3: The symbol h should be defined.
- Line 294: The dimensions of 14.30, 21.92 and 30.91 should be given (L/s).
- See annotated manuscript! (the letters and numbers in many figures are too small).

Round 3
Reviewer 4 Report
- Most comments of my last review were taken into account by the authors in the revised version of the manuscript.
- The comments that were not taken into account by the authors in the revised version, are given on the annotated manuscript.
